# Non-Invasive Imaging Methods to Evaluate Non-Alcoholic Fatty Liver Disease with Fat Quantification: A Review

**DOI:** 10.3390/diagnostics13111852

**Published:** 2023-05-25

**Authors:** Weon Jang, Ji Soo Song

**Affiliations:** 1Department of Radiology, Jeonbuk National University Medical School and Hospital, 20 Geonji-ro, Deokjin-gu, Jeonju 54907, Jeonbuk, Republic of Korea; weon0315@gmail.com; 2Research Institute of Clinical Medicine, Jeonbuk National University, Jeonju 54907, Jeonbuk, Republic of Korea; 3Biomedical Research Institute, Jeonbuk National University Hospital, Jeonju 54907, Jeonbuk, Republic of Korea

**Keywords:** hepatic steatosis, non-alcoholic fatty liver disease, non-invasive quantitative biomarker, liver fat quantification

## Abstract

Hepatic steatosis without specific causes (e.g., viral infection, alcohol abuse, etc.) is called non-alcoholic fatty liver disease (NAFLD), which ranges from non-alcoholic fatty liver (NAFL) to non-alcoholic steatohepatitis (NASH), fibrosis, and NASH-related cirrhosis. Despite the usefulness of the standard grading system, liver biopsy has several limitations. In addition, patient acceptability and intra- and inter-observer reproducibility are also concerns. Due to the prevalence of NAFLD and limitations of liver biopsies, non-invasive imaging methods such as ultrasonography (US), computed tomography (CT), and magnetic resonance imaging (MRI) that can reliably diagnose hepatic steatosis have developed rapidly. US is widely available and radiation-free but cannot examine the entire liver. CT is readily available and helpful for detection and risk classification, significantly when analyzed using artificial intelligence; however, it exposes users to radiation. Although expensive and time-consuming, MRI can measure liver fat percentage with magnetic resonance imaging proton density fat fraction (MRI-PDFF). Specifically, chemical shift-encoded (CSE)-MRI is the best imaging indicator for early liver fat detection. The purpose of this review is to provide an overview of each imaging modality with an emphasis on the recent progress and current status of liver fat quantification.

## 1. Introduction

Non-alcoholic fatty liver disease (NAFLD) is a disease spectrum characterized by hepatic steatosis in the absence of specific causes such as viral hepatitis, alcohol abuse, steatogenic medications, or genetic lipodystrophies. This is currently the most prevalent chronic liver disease in the world, with an approximate worldwide prevalence of 25%. NAFLD ranges in its severity from non-alcoholic fatty liver (NAFL), which is an isolated steatosis, to non-alcoholic steatohepatitis (NASH), fibrosis, and NASH-related cirrhosis, which is on the more severe end of the spectrum [1,2].

Beside steatosis, NASH involves necro-inflammatory alteration to the hepatocytes, showcasing the disease’s progressive nature [3,4]. Approximately 33% of NAFL and NASH patients progress to inflammation, hepatocyte injury, and fibrosis, although about 20% can exhibit some disease regression. Hepatic decompensation and cirrhosis develop over a mean of 7.6 years in an estimated 3% of patients with NAFLD. Furthermore, patients with decompensated NASH have an average survival of only 2 years [5]. Moreover, a recent meta-analysis showed that moderate to severe hepatic steatosis in NAFLD patients were closely related to clinically significant coronary artery disease [6].

Hepatic steatosis is the histopathological indicator of NAFLD, but it can exist in many additional pathologies. Patients with hepatic steatosis carry a risk of detrimental complications, such as fibrosis, steatohepatitis, end-stage liver disease, and hepatocellular carcinoma [3,4,7]. A recent large cohort study showed that hepatic steatosis could be an independent predictor of mortality at a population level, as well as hepatic fibrosis [8]. In addition, 30% reduction in liver fat deposition based on magnetic resonance imaging proton density fat fraction (MRI-PDFF) can be used to estimate the odds of fibrosis regression in NAFLD [9]. However, there are a significant number of patients with NAFLD at early stages, more than can justify medical treatment for all patients. Instead, the large prevalence of severe later complications indicates the necessity for preventive measures in patients with more advanced NAFLD.

Currently, nontargeted liver biopsy remains the gold standard for the diagnosis of NAFLD. Histologically, hepatic steatosis is graded based on the proportion of hepatocytes within the intracellular lipid-containing vacuoles and is divided into four grades (normal, mild, moderate, and severe) depending on how much it affects hepatocytes. Based on the work of Brunt et al. [10], grade zero (<5% hepatocytes affected, S0) is considered normal, grade one (5–33% hepatocytes affected, S1) is considered mild, grade two (34–66% hepatocytes affected, S2) is considered moderate, and grade three (>66% hepatocytes affected, S3) is considered severe. Generally, a common threshold for moderate steatosis has been 30% [11,12]. Liver biopsy is also performed to determine whether inflammation and significant fibrosis are present. Particularly, liver biopsy is still needed for the conclusive diagnosis of NASH [13], and NAFLD-risk stratification requires distinguishing patients with inflammation and/or fibrosis, i.e., distinguishing patients with NASH from those patients with isolated steatosis [5].

A crucial limitation of liver biopsy is its impracticality in routine and repeated steatosis assessment due to high cost, sampling inaccuracy, and consequences related to its invasiveness such as pain, infection, or bleeding. In addition, limited patient acceptance and low intra- and inter-observer repeatability have also been raised [14]. Importantly, the histopathologic features of NAFLD are patchy at the spatial scale of a biopsy. As a result of this heterogeneous distribution of fat in the liver, variability due to sampling error frequently occurs [15,16].

For the above mentioned reasons, there is a substantial motivation for the development of non-invasive approaches for the management of NAFLD, including predictive models such as NAFLD fibrosis score, serum biomarkers (enhanced liver fibrosis test (ELFTM)), and imaging techniques that quantify hepatic steatosis or liver stiffness [10], and assess other potential quantitative imaging biomarkers. Thanks to the recent progress in modern imaging techniques over the past 2 decades, non-invasive imaging techniques that can detect and quantify hepatic steatosis are increasingly chosen in clinical settings [17].

This article focuses on non-invasive imaging techniques such as ultrasound (US), computed tomography (CT), and magnetic resonance imaging (MRI) for the diagnosis and staging of NAFLD, providing an overview of the concepts, diagnostic performance, and advantages and limitations of each approach.

## 2. Ultrasonography

US has several advantages over other imaging modalities such as MRI and CT. US is widely available worldwide, and there is no radiation exposure. As a result, US has been standardly used as the key imaging modality in evaluating hepatic steatosis. The echo change in hepatic parenchyma is the basis in US for evaluation of hepatic steatosis. Hepatic steatosis appears as a diffuse increase in hepatic parenchymal echogenicity, also known as a “bright liver” on US, as the intracellular accumulation of fat vacuoles reflects the ultrasound beam [18].

### 2.1. Conventional US

Conventional US (B-mode US) is the modality of choice for the initial examination of high liver enzymes, which are frequently associated with hepatic steatosis. However, most US examinations are qualitative and restricted in their performance, especially in overweight individuals at high risk of NAFLD [19].

B-mode US allows an estimate of steatosis severity according to the subjective examination of sonographic patterns [20]. The severity is generally categorized as absent, mild, moderate, and severe. Mild hepatic steatosis is seen as a more diffuse increase in liver echogenicity than the renal cortex, moderate hepatic steatosis is seen as an increase in liver echogenicity with impaired visibility of the diaphragm and portal vein wall, and severe hepatic steatosis is seen as a large increase in liver echogenicity and poor visualization of the diaphragm, portal vein wall, and posterior regions of the right liver lobe (Figure 1).

US can offer reasonable accuracy in detecting moderate to severe hepatic steatosis with a sensitivity and specificity of about 90% and 95%, respectively, for patients lacking concomitant chronic liver disease [18,20]. Nevertheless, it has a mere moderate diagnostic ability in detecting hepatic steatosis of 5% or greater (mild steatosis; sensitivity, 50–62%). Further, it may not succeed in overweight patients or patients with ascites; the modality is strongly operator- and platform-dependent [21].

Conventional US carry disadvantages that are mostly due to inter-observer bias, though the technique can be made more objective in many ways, most straightforwardly, by normalizing the signal to the cortex of a healthy kidney. The liver-to-kidney ratio (hepatorenal index) (Figure 2) can be used to improve the technique and it has been useful in detecting liver steatosis [22,23]. However, the hepatorenal index has limitations in some patients as a result of the comorbid existence of kidney disease and anatomical variations, and it has been shown to struggle between differentiating absent and mild steatosis [24].

### 2.2. Quantitative US

While traditional US can be used for many medical situations, the quantitative information derived from B-mode US can be limited due to the images being strongly dependent on machine settings. That said, modern developments can allow ultrasound scanners to not only provide images but also to provide radiofrequency (RF) data that can enable quantitative ultrasound (QUS) [25,26].

The three quantitative parameters of US investigated by the initiative are backscatter coefficient (BSC), attenuation, and speed of sound (SoS). Ultrasound waves diminish in energy when they go through the liver. Specifically, they lose greater energy as they pass through fatty liver tissue than through the normal liver, which results in a larger attenuation coefficient. Scattering happens when a wave interacts with the microstructure of liver tissue. Greater scattering occurs in fatty liver tissue compared to normal liver tissue, which will result in a greater backscatter coefficient. As a result, the ultrasound wave speed is reduced in fatty liver tissue compared to normal liver tissue [27]. These methods analyze the radiofrequency echoes that return to the transducer and calculate parameters that may be used to determine the amount of fat in the liver [28]. Lately, non-invasive quantitative US has been used more frequently in order to reduce patient risk and offer quick results as well as a MRI-PDFF [29].

### 2.3. Attenuation Coefficient

Among the above-mentioned techniques, the attenuation coefficient (AC) has been more widely studied [27]. Ultrasound attenuation is greater with hepatic fat infiltration, which hides the hepatic vessels and diaphragm in conventional US [30].

Attenuation is the energy loss that occurs when an ultrasound wave passes through tissue, and this energy loss results in a reduced signal return to the transducer, which appears as hypoechoic regions in deep tissues. It depends on the tissue properties and the US frequency, so that the presence of fat in a tissue increases the signal delay by increasing attenuation [25,28].

The AC is a quantitative measurement of energy loss during US transmission [26]. There are two central methods to evaluate hepatic steatosis using AC. The first is to use a controlled attenuation parameter (CAP) obtained with the transient elastography device using A-mode ultrasound. The second is to use B-mode US-guided attenuation imaging.

#### 2.3.1. Controlled Attenuation Parameter

The CAP consumes less time and provides an evaluation of steatosis and fibrosis simultaneously [31]. It is expected to be observer-independent with a good inter-observer agreement (concordance correlation coefficient, 0.82 between two raters) [32]. However, the CAP can be changed by various factors, including skin to capsule distance [33] and probe type (M vs. XL probe) [34] and the cutoff value for the diagnosis of hepatic steatosis is not standardized well and varies across studies [27]. Additionally, CAP measurement from a sample volume is blindly recorded without a B-mode ultrasound image; as a result, the CAP value can be evaluated incorrectly as a result of mistaken inclusion of ducts, masses, hepatic vessels, or uneven steatosis [35].

#### 2.3.2. B-Mode Ultrasound-Guided Attenuation Imaging

New techniques for calculating the AC under B-mode ultrasound guidance have been introduced to evaluate hepatic steatosis, such as attenuation imaging (ATI) [36,37], ultrasound-guided attenuation parameter (UGAP) [38], attenuation coefficient (ATT) [39], and tissue attenuation imaging (TAI) [40].

While the details of the evaluation method differ slightly between companies, the typical measurement process is as follows: (1) A convex probe is used to perform a B-mode US evaluation of the liver; (2) the probe is used to visualize the right hepatic lobe via an intercostal window for AC measurement; (3) the region of interest (ROI) is fixed in the right hepatic lobe at least 2 cm below the liver capsule to mitigate reverberation artifacts during breath-hold while excluding large vessels; and (4) the AC value (in dB/cm/MHz) and reliability of the measurement (in R2) are determined (Figure 3). A measurement of R2 ≥ 0.60–0.90 is considered satisfactory, and approximately five satisfactory measurements are used to assess hepatic steatosis. The technical failure rate for these methods, including ATI and UGAP, appears low (0–4.3%), although there is limited data available [36,38,39,40,41,42].

In various studies published recently, the AC calculated with these methods usually demonstrated a positive diagnostic performance for hepatic steatosis, with liver biopsy or MRI-PDFF as a reference standard [36,38,39,41,43,44]. The benefit of these methods over CAP is their employment of B-mode US images. Firstly, conventional US evaluation of the liver can be done at the same time as fat quantification. Secondly, the ROI for determining the AC can be placed during visualization of the liver, and a more dependable outcome can be determined by avoiding large vessels, ducts, and hepatic mass [36,38,39]. In a recent study, a quantitative US fat fraction estimator utilizing seven parameters (attenuation coefficient, backscatter coefficient, Lizzi-Feleppa slope, intercept, midband fit, and envelope statistics parameters *k* and µ) showed better accuracy than CAP in the diagnosis of hepatic steatosis in NAFLD patients [45].

#### 2.3.3. Miscellaneous

The BSC is a quantitative measurement of ultrasound energy reflected from tissue. It is associated with the echogenicity of the tissue in conventional US. As echogenicity increases with hepatic steatosis in conventional US, the BSC also increases with hepatic fat infiltration [26,46]. The BSC has a strong diagnostic performance in detecting hepatic steatosis (AUROC, 0.85 and 0.83 for ≥S2 and ≥S3 and 0.95 for MRI-PDFF ≥ 5%) [26,47], with MRI-PDFF or a biopsy as standards of reference. Nevertheless, these were research studies that necessitated post-processing of the QUS data.

SoS measurement is a QUS parameter that can be used to determine tissue properties associated with changes in ultrasound echo wave speeds in different media [48]. SoS has been shown to decrease in proportion with an increase in liver fat content [49]. A previous study showed strong results in distinguishing hepatic steatosis for SoS compared to MRI-PDFF and biopsy (AUC of 0.942 and 0.952, respectively) [50]. Larger studies on patients with suspected or known NAFLD are warranted to confirm the clinical applicability of BSC and SoS.

### 2.4. Limitation

Since the acoustic characteristics of hepatic tissue alter with hepatic fat accumulation, it is anticipated that QUS will be able to detect hepatic steatosis with improvement in accuracy and reproducibility and to quantitatively evaluate hepatic steatosis [35]. However, due to the difficulty of harmonizing methodologies across vendors and platforms, the availability of various developing quantitative US approaches from many vendors may slow down distribution.

On the other side, inflammation and fibrosis are key histologic features of NAFLD, which can alter the treatment strategy [51]. While there was a study evaluating lobular inflammation of the liver in NAFLD patients [52] that showed promising results using a shear-wave dispersion slope acquired with two-dimensional shear-wave elastography, further studies with large numbers of patients from various countries are warranted. The outcome may be useful in enabling a comprehensive evaluation of patients with NASH or NAFLD utilizing ultrasonography with QUS techniques for hepatic fat quantification.

## 3. Computed Tomography

CT is a generally used imaging technique for the abdominal exam that can objectively quantify liver fat content. X-ray absorption of the fatty tissue is less than that of normal hepatic tissue, resulting in a decrease in attenuation as fat concentration increases [11,53].

### 3.1. Conventional Unenhanced CT

Normal liver parenchyma is about 60 HU in unenhanced CT, and it hyperattenuates relative to the spleen [54], while steatosis is approximately at 40 HU, and the liver tissue hypoattenuates relative to the fat-free spleen [55] (Figure 4). The sensitivity and specificity of unenhanced CT for low-grade steatosis (cut-off values, 10–20%) are 57% and 88%, respectively. For high-grade steatosis (cut-off values, 25%), the sensitivity and specificity increase to 72% and 95%, respectively [21]. A HU threshold of 48 in unenhanced CT acquired at 120 kVp has been demonstrated to be strongly specific (100%) for high-grade steatosis (~30%), with a positive predictive value of 100%, negative predictive value of 94%, and a sensitivity of 54% [56].

Unenhanced CT is usually preferable to contrast-enhanced CT for predicting pathologic liver fat content as assessed by histopathology because iodine-based contrast agents increase hepatic attenuation, sometimes preventing precise quantification of liver fat content [55,56]. However, the absolute attenuation value of liver parenchyma on an unenhanced CT scan can be affected by beam hardening effects in patient with a large body habitus and CT acquisition parameters including kVp and vender-specific filters and reconstruction algorithms [57]. Therefore, instead of using an absolute attenuation value of liver parenchyma on an unenhanced CT scan (CT_L_), attenuation differences between the liver and spleen on unenhanced CT using the spleen as an internal control have been thought to be a more adequate quantitative parameter to evaluate hepatic steatosis [18].

A recent large cohort study showed that both liver and spleen attenuation difference (CT_L-S_) and ratio (CT_L/S_) performed better and were less dependent on the CT technique for diagnosing hepatic steatosis than liver attenuation alone, and CT_L-S_ had the highest diagnostic performance among the three mentioned CT indices [58].

Clinical CT provides a significant potential for detecting incidental steatosis and may aid in clarifying the standard course of NAFLD [59]. That said, it is essential to be aware of certain confounding factors such as superimposed iron, iodine contrast, and hepatitis [55,60]. These circumstances increase liver attenuation and may appear like or mask the already existing steatosis. Notably, the consequence of iron on CT attenuation is negligible and may only be significant in severe or moderate iron overload conditions [59].

### 3.2. Dual Energy CT

Dual energy CT (DECT), which uses two separate energy levels, may differentiate between many chemical compositions inside tissues including fat, and hence has the potential to provide superior diagnostic performance for detecting hepatic steatosis compared to traditional single energy CT [60,61]. However, it has no clear advantage over simple attenuation measurements with unenhanced images [62]. Nevertheless, some studies show that attenuation measured at virtual non-contrast (VNC) CT is moderately correlated with liver fat content and has >90% specificity for the diagnosis of hepatic steatosis [63]. The liver attenuation index of VNC was significantly correlated with that of NECT image and might be feasible for diagnosing substantial hepatic steatosis in living liver donor candidates using different cutoff values of the liver attenuation index such as CT_L-S_ [64]. The most likely function of dual-energy CT is to distinguish between superimposed iodine or iron overload [62,65,66], which can obscure the presence of steatosis in single-energy CT. With multiple material decomposition (MMD) algorithms, which can be used to decompose DECT data into multiple materials, it can reliably detect and grade hepatic steatosis. MMD algorithm-derived fat images show values as the percentage fat fraction, providing a quantitative assessment of liver steatosis [61,67].

### 3.3. Photon Counting CT (PCCT)

Recently, as a further advancement of spectral CT technology, photon-counting CT (PCCT), which is the first clinical CT scanner utilizing a photon-counting detector (PCD), was developed. Conventional CT detectors, also known as energy-integrating detectors (EID), incidentally absorb X-rays converted into visible light in the upper layer made of a scintillator. The amount of light is measured and converted to an electrical signal proportional to the total energy deposited during a measurement interval (Figure 5A).

On the other hand, PCDs differ from EID in that they consist of a single thick sheet of a semiconductor material and can detect individual photons and their associated energies. When an incident X-ray is absorbed in a semiconductor, a cloud of positive and negative charges is generated. An electrical pulse is created in the wires attached to the electrodes, which is then registered using an electronic readout circuit. PCDs therefore convert single X-ray photons collectively into an electric signal [68,69] (Figure 5B). This signal can be used to detect and quantify several materials within the body, such as fat and iodine. Therefore, this method is useful for precise measurement in monitoring hepatic steatosis [70].

A recent study compared liver fat quantification between MRI-derived fat fraction and unenhanced PCCT-derived fat fraction using phantom and obese adult patients and showed promising results. Although this was a pilot study, the results showed high accuracy in liver fat fraction quantification for PCCT compared to the current clinical standard of MRI (average difference in fat fraction, 1.1 ± 1.9%, *p* = 0.138) [70]. Another study using VNC acquired from spectral datasets with PCCT showed excellent sensitivity, as well as specificity in the detection of hepatic steatosis (sensitivity, specificity, PPV, and NPV were 94%, 92%, 41%, and 99.6% for CT_L_, 96%, 90%, 46% and 99.6% for CT_L-S_ and 95%, 99.6%, 42%, and 99.6% for CT_L/S_) [71].

### 3.4. Future Directions

As deep learning based on artificial intelligence develops, automatic assessment of CT liver HU values has been developed for liver fat quantification [72]. Complete automatic liver segmentation can improve reproducibility and objectivity, which can avoid human bias, and can show promising results with a high correlation between automatic and manual HU and MRI-PDFF measurements [72,73]. Although further advances are still required, automated algorithms show promise for rapid and objective measurement of liver fat.

Since ionizing radiation is a big hurdle for the use of CT as a diagnostic tool for quantifying hepatic steatosis, utilizing VNC acquired from PCCT on contrast-enhanced CT might be promising. However, large prospective studies using MRI-PDFF or pathology as a reference standard are needed. In addition, it is anticipated that further research discriminating the level of inflammation that is present in NAFLD is required.

## 4. Magnetic Resonance Imaging

MRI provides a large array of methods to detect and quantify liver fat content through detecting proton signals present in water and fat [74]. Hepatic steatosis assessment has improved from conventional MRI methods from qualitative estimates to quantitative MR spectroscopy (MRS) and MRI methods, which allow more accurate measurements of liver fat content [17,74].

### 4.1. Conventional Qualitative MRI

Conventional qualitative methods, including chemical shift image (Figure 6) or fat-suppression techniques (T1-weighted gradient-echo and T2-weighted fast spin-echo sequences), have been used for qualitative evaluation of steatosis. However, they are unsuitable for quantitative evaluation due to multiple confounding variables that reduce their accuracy (discussed later).

### 4.2. Quantitative MRI

Triglycerides found in tissue may be quantitatively assessed with confounder-corrected chemical shift-encoded (CSE)-MRI and confounder-corrected MRS. CSE-MRI and MRS both take advantage of a chemical shift in resonance frequencies between fat and water. A reduction in the electronic shielding of protons in water molecules, compared to protons in triglycerides, can lead to an increased resonance frequency of water compared to fat by 3.4 ppm, since the difference between main methylene resonance and water is highest at body temperature. Both MRS- and CSE-MRI take advantage of this “chemical shift” to distinguish fat and water proton signals [74].

When proton MRI signals of fat and water are distinguished and measured, a normalized fat signal ratio is determined. When the signals are in proportion with the proton density of fat and water, the consequent ratio is the same as the proton density fat fraction (PDFF), which is defined as follows: PDFF = F/(W + F), where F and W are unconfounded signals from protons inside mobile triglycerides and water molecules [75,76]. The PDFF is a percentage (range, 0–100%), and it is associated with the fat percentage at histologic examination [21,76]. The PDFF values are classified into four grades: S0 is absent (<5.5%), S1 is mild (5.5–16.2%), S2 is moderate (16.3–21.6%), and S3 is severe (>21.7%) [77]. MRI- PDFF has been increasingly adopted as the best technique among all methods for quantifying liver fat content, as it even overperforms a biopsy [29,78].

In addition, MRI-PDFF has a high degree of reproducibility across different imaging manufacturers, field strengths, and imaging centers [79]. This is key for PDFF standardization as a clinically recognized biomarker and the general dissemination of information regarding its use. Even though PDFF and fat percentage assessed with a histopathologic examination are not interchangeable, there exists a good correlation between the two [29].

Details on confounding factors, advantages and disadvantages, and diagnostic performances of MRS- and CSE-MRI will be discussed in the following section.

#### 4.2.1. Magnetic Resonance Spectroscopy

MRS enables depiction of the proton signals of water and fat as separate peaks in a high-resolution spectrum from a single voxel acquired during a single breath-hold (approximately 15–20 s) [80]. Confounders of the MRI signal for MRS includes T1-related bias, T2 decay, and J-coupling [74].

For the liver parenchyma, most of the protons are within water and fat, and consequently, most of the identifiable peaks on MRS of the liver come from water and fat protons. The water proton peak is seen as a single peak at 4.7 ppm, and fat protons are seen as multiple peaks due to chemical bonds between protons and adjacent atoms in fat. Therefore, fat detection with MRS can be straightforward in theory, as it requires only the detection of spectral peaks at certain frequencies that correspond to fat protons [81].

The methods used most often are stimulated-echo acquisition mode (STEAM) and point-resolved spectroscopy (PRESS). PRESS has a greater signal-to-noise ratio (SNR) than STEAM. However, STEAM shows fewer effects from J-coupling, and it is often preferred [82]. Since histopathologic examination is the standard of reference, a recent meta-analysis demonstrated confounder-corrected MRS to have a sensitivity of 73–89% (compared with 73–91% for US and 82–97% for CT) and a specificity of 92–96% (compared with 70–85% for US and 88–95% for CT) [21].

Well-performed single-voxel MRS provides high intra- and inter-examination repeatability [83]. Moreover, several studies reported that the PDFF obtained from MRS was better correlated with actual fat content compared to the histopathological assessment of hepatic steatosis performed by pathologists [84,85,86]. Nevertheless, a small voxel (approximately 2–9 cm^3^) has the same issue in biopsy sampling variability. Multivoxel MRS can be exploited to range over larger volumes, but this increases the scanning time as a result (from minutes to hours, depending on encoding techniques and area covered) [87]. Moreover, due to the limited tissue sampling of MRS, when comparing volumetric CSE-MRI methods, retesting variance in MRS is often seen as being larger for MRS than it is for CSE-MRI.

#### 4.2.2. Chemical Shift-Encoded MRI

Chemical shift-encoded (CSE)-MRI separates fat and water signal components by recording spoiled gradient echoes at various echo times, often in one repetition [88]. CSE-MRI may be performed in a single breath-hold and may offer an almost real-time PDFF map reconstruction across the whole liver in about 15–20 s [76] (Figure 7).

To calculate the PDFF, MRI data should be postprocessed and reconstructed through fitting to an accurate spectral model of water and fat that is corrected for confounders, such as T1-related bias, T2* decay, eddy currents, spectral complexity of fat, noise-related bias, and concomitant gradients. Magnitude- and complex-based strategies are standardly employed in the postprocess and determination of CSE-MRI-PDFF [76].

Magnitude-based CSE-MRI employs the gradient-echo signal magnitude rather than the signal phase. It is easier to use, and it is more resistant to errors in phase caused by things like B0 field inhomogeneities and eddy currents [89,90]. That said, magnitude-based methods have a reduced SNR and a restricted PDFF dynamic range of 0–50%. On the other side, complex-based CSE-MRI makes use of both the phase and magnitude components of the signal and may allow the entire PDFF range to be measured (0–100%). Complex-based CSE-MRI has a greater SNR, but it is more sensitive to errors in phase and may have problems with water-fat swaps in inhomogeneous B0 fields [91,92].

Since it is unusual for PDFF to be larger than 50%, the dynamic range of magnitude-based CSE-MRI is meaningful only in situations besides the liver (e.g., adipose tissue and bone marrow). To overcome this limitation, hybrid approaches have been created that combine the phase insensitivity of magnitude-based techniques with the high SNR and complete dynamic range of complicated methods [90]. Whether CSE reconstruction is complex-based, magnitude-based, or a combination of both (the hybrid method) is dependent on the company.

CSE-MRI-PDFF has a linear correlation with MRS-PDFF (*R*^2^ = 0.96), and it has a coefficient of repeatability and reproducibility of 3.0% and 4.1%, respectively [78]. In fact, the ability to investigate the liver in its entirety avoids the sample restrictions of both liver biopsy and MRS. When it comes to longitudinal investigations or treatment monitoring, this is a key advantage, especially when various sites or scanner platforms are involved. Exact PDFF measurement co-localization from volumetric data sets recorded at multiple time intervals is made possible by sampling the entire liver.

In addition to evaluating hepatic steatosis, CSE-MRI allows simultaneous assessment of iron deposition. Iron is a paramagnetic substance and consequently reduces the T2* (i.e., increases the relaxation rate R2* as R2* = 1/T2*) and results in signal loss with an increase in echo time. The coexistence of fat and iron is relevant in conditions such as hepatocellular carcinoma, hemochromatosis, viral hepatitis, and hemosiderosis [93]. However, concomitant liver steatosis may hide R2* signal decay in the liver and confound the interpretation of iron deposition. With simultaneous estimation of PDFF and R2*, CSE-MRI offers fat-corrected R2* maps that allow liver quantification independently of the existence of fat [74,93]. Unlike other imaging methods such as conventional US or CT, CSE-MRI is therefore a useful imaging modality for fat and iron deposition that coexists in the liver. The characteristics, advantages, and limitations of conventional chemical shift images, MRS, and CSE-MRI are briefly shown in Table 1.

### 4.3. Future Directions

#### 4.3.1. Automated Measurement

Currently, measuring fat fraction and iron concentration require manually selecting ROIs in the liver parenchyma. This strategy consumes time since multiple ROIs are needed to obtain satisfactory results [94]. The diagnostic performance of PDFF manual whole-liver segmentation (WLS) demonstrates a good correlation with results using alternative manual ROI sampling methods and spectroscopy [95,96]. Consistent with repeatability conditions, a similar inter-observer agreement can be obtained using manual ROIs and WLSs, both of which contribute to the variability in radiomic measurements. Furthermore, automated convolutional neural network (CNN) solutions have evolved to track WLSs with diagnostic accuracy that is similar to manual segmentation [73]. Moreover, analysis automation can demonstrate the feasibility of large data collection and analysis [97]. Considering there is no effective approach to evaluate the level of inflammation in NAFLD using MRI, these directions for development are anticipated to aid in the diagnosis of progressing disorders such as NASH.

Automated methods can give results that are objective and limit bias introduced by humans. However, manual interaction still might be required when full automation is difficult. Further evolution in automated PDFF analysis methods is still needed.

#### 4.3.2. Reduction of Acquisition Time

On the basis of recent developments, alternative advanced techniques for time shortening have emerged such as compressed sensing and MR fingerprinting for assessing fat quantification of the liver [98,99,100]. Despite requiring a pulse sequence from high-performance hardware and software [101], compressed sensing MRI reconstruction enables reconstruction of MRI scans by acquiring fewer data via variable density incoherent undersampling of k-spaces, hence reducing acquisition time [98].

MR fingerprinting sequences reveal quantitative multi-parametric liver tissue characterization in a single breath-hold scan with measurements of properties of tissues and relaxation parameters. There are a variety of MRI settings and parameters in fingerprinting during data acquisition, which create specific signal patterns. The “fingerprints” are then matched to samples from a dictionary of signal patterns created with Bloch equation simulations. After they are matched, tissue properties used to generate the fingerprint are determined as pixel-wise maps [100].

## 5. Conclusions

Hepatic steatosis is regarded as a distinct risk factor for the progression into NASH in NAFLD, which may result in both hepatic and extrahepatic disorders. The increasing frequency of NAFLD and acknowledgment of the disease’s burden, along with the limits of liver biopsies, have led to the rapid development of non-invasive imaging techniques capable of objectively assessing hepatic steatosis.

Several imaging techniques and strategies of US, CT, and MRI for quantitative evaluation of hepatic steatosis have been investigated and improved. Each method has benefits and drawbacks. US can be conducted efficiently at the bedside without exposing the patient to radiation, with recent improvement in diagnostic accuracy of fat quantification; however, examinations of the entire liver are limited. CT is readily available and useful for detection and risk classification, especially when evaluated using artificial intelligence, but it poses a radiation exposure and requires further refinement and evaluation to be adopted clinically. Although MRI is costly and has a relatively long acquisition time, it can provide information regarding the percentage of liver fat (MRI-PDFF) and has potential to predict the odds of fibrosis regression. Particularly, CSE-MRI has been shown to be the most accurate imaging biomarker for identifying and quantifying liver fat levels and has emerged as the primary tool for liver fat quantification. This precise and reproducible measurement of hepatic steatosis has the potential to revolutionize the design of diagnostic techniques and treatment trials for NAFLD.

## Figures and Tables

**Figure 1 diagnostics-13-01852-f001:**
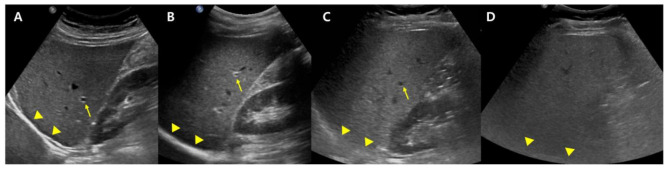
Conventional US examination for evaluation of hepatic steatosis: normal (**A**), mild (**B**), moderate (**C**), and severe (**D**) hepatic steatosis. The vessel (arrow) and diaphragm (arrowhead) are well distinguishable in a normal liver (**A**). Liver parenchymal echogenicity is increased compared with renal cortical echogenicity in mild hepatic steatosis (**B**). In moderate hepatic steatosis (**C**), the vessel wall echo becomes obscured (arrow), and the diaphragm is partially visible (arrowhead). Due to a marked increase in liver echogenicity in severe hepatic steatosis (**D**), there is blurring and poor visualization of the diaphragm (arrowhead), as well as deep posterior parts of the right liver lobe.

**Figure 2 diagnostics-13-01852-f002:**
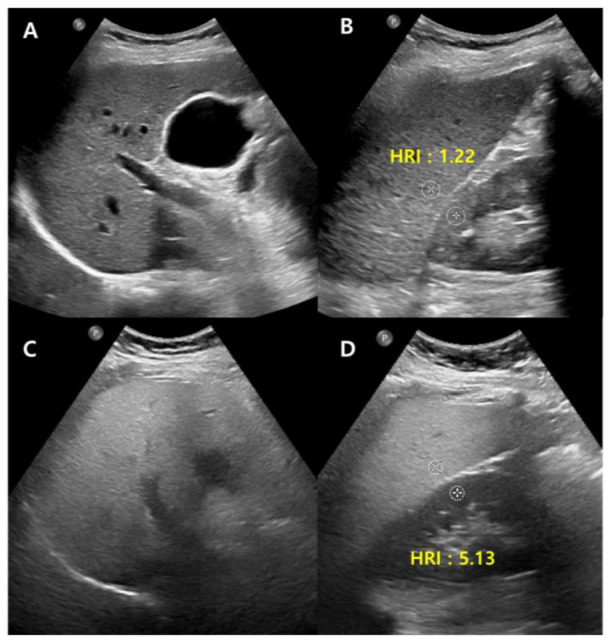
Conventional US of a healthy liver (**A**) and severe hepatic steatosis (**C**) with hepatorenal index (HRI) (**B**,**D**). Normal hepatic parenchyma echogenicity shows well-visible intrahepatic structures such as the portal vein, hepatic vein, and liver parenchyma (**A**). The HRI measures 1.22 (35.1/28.8, normal < 1.5) in a healthy liver (**B**). Severe hepatic steatosis shows poorly visible portal vein wall echo and hepatic vein (**C**), and the HRI measures 5.13 (66.8/13.0) (**D**).

**Figure 3 diagnostics-13-01852-f003:**
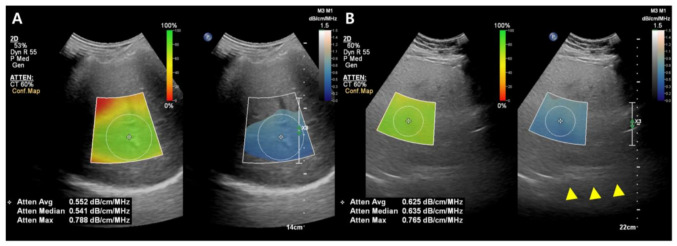
Images from the attenuation coefficient (AC) from the Philips system (**A**,**B**). Mean AC value of image (**B**) (0.625 dB/cm/MHz) is higher than image (**A**) (0.552 dB/cm/MHz) which means a larger fat component in the image (**B**). The echo of the diaphragm is poorly visible (arrowhead). Confidence map is shown as a color box, and the poor-quality areas are not included in the measurement.

**Figure 4 diagnostics-13-01852-f004:**
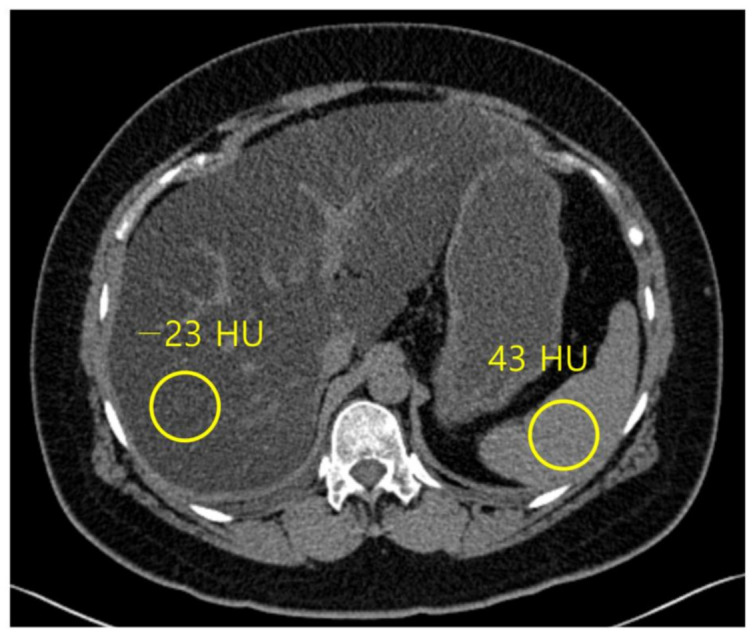
Non-contrast computed tomography shows the attenuation of the liver and spleen. The mean attenuation of the liver (−23 HU) is remarkably lower than the mean attenuation of the spleen (43 HU), which represents severe hepatic steatosis.

**Figure 5 diagnostics-13-01852-f005:**
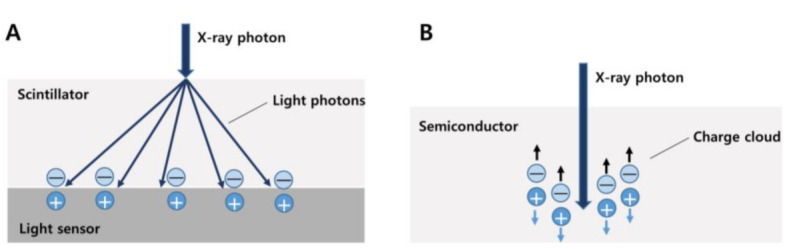
Different detector types of conventional CT (**A**) and photon-counting CT (**B**). In a conventional energy-integrating detector, an incident X-ray photon is converted into a shower of visible light photons in a scintillator. Visible light hits an underlying light sensor, where it produces both positive and negative electrical charges (**A**). In a photon-counting detector, the X-ray photon is absorbed in a semiconductor material, where it produces both positive and negative electrical charges. Under the influence of a strong electric field, the positive and negative charges are pulled in opposite directions, generating an electrical signal (**B**).

**Figure 6 diagnostics-13-01852-f006:**
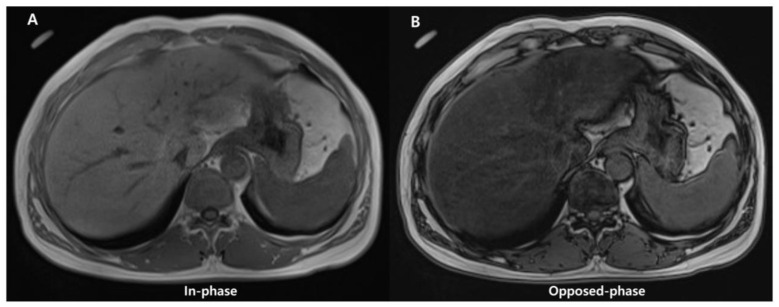
Images of in- (**A**) and opposed-phase (**B**) of chemical shift MRI. In (**B**), the liver parenchymal signal is lower compared with (**A**) due to signal drop caused by hepatic fat deposition.

**Figure 7 diagnostics-13-01852-f007:**
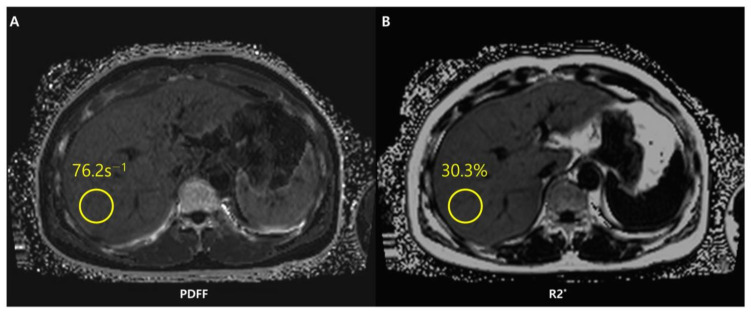
Chemical shift-encoded (CSE)-MRI allows an estimate of both fat in the liver and iron deposition simultaneously. Fat-corrected R2* mapping (**A**) is a standard byproduct of multi-echo CSE acquisitions used to map R2*-corrected proton density fat fraction (PDFF) (**B**). The system has detected a PDFF value of the ROI equal to 30.3%. The patient thus presents severe steatosis.

**Table 1 diagnostics-13-01852-t001:** The characteristics, advantages, and limitations of conventional chemical shift images, MRS, and CSE-MRI.

Title 1	Characteristics	Advantages	Limitations
Conventional chemical shift image	Generated from in-phase opposed-phase imaging which exploits echo time-dependent phase interference effect between gradient echo signals of water and fat	Enables qualitative assessment of hepatic steatosis intuitively	Unsuitable for quantitative evaluation due to multiple confounding variables
MRS	Proton signals of water and fat obtained from a single voxel depicted as separate peaks in a high-resolution spectrum	Measures PDFF, with high intra- and inter-examination repeatability	Sampling variability due to use of a single voxel
		Enable multiple-points measurement	Increased image acquisition time when covering larger area
CSE-MRI	Separate water and fat signal components by means of strategically sampling spoiled gradient echoes at multiple echo times (typically six echoes)	Measures PDFF, with high intra- and inter-examination repeatability	Limited PDFF dynamic range (0–50%) in magnitude-based method
		Enables sampling of the entire liver from volumetric data set	

## Data Availability

No new data were created or analyzed in this study. Data sharing is not applicable to this article.

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
