# Peer review of "Non-Invasive Imaging Methods to Evaluate Non-Alcoholic Fatty Liver Disease with Fat Quantification: A Review"

_diagnostics, 2023, doi:10.3390/diagnostics13111852_

Round 1
Reviewer 1 Report
Dear authors,
I want to congratulate you on this topic.
Non-alcoholic fatty liver disease (NAFLD) is now a major emerging health problem and the most common cause of chronic liver disease in many developed countries. Traditionally, liver biopsy has been the gold standard method for quantifying hepatic steatosis. But now, different imaging studies have been used to evaluate hepatic steatosis.
You have carried out an almost complete review of the non-invasive techniques for quantifying hepatic steatosis. Transient elastography and acoustic radiation force impulse (ARFI) elastography are also important tools used to determine the degree of liver stiffness. Perhaps it would have been useful to develop a little more the contribution of these techniques to the diagnosis of non-alcoholic fatty liver disease.
It is a article with many references. Maybe too synthetic, but written for the understanding of any professional, from GP to gastroenterologist specialist.
Good luck!
Dear Editor,
First of all I want to thank you for the proposal to revise this article.
Non-alcoholic fatty liver disease (NAFLD) is now a major emerging health problem and the most common cause of chronic liver disease in many developed countries. Traditionally, liver biopsy has been the gold standard method for quantifying hepatic steatosis. But now, different imaging studies have been used to evaluate hepatic steatosis.
The authors have carried out an almost complete review of the non-invasive techniques for quantifying hepatic steatosis. Transient elastography and acoustic radiation force impulse (ARFI) elastography are also important tools used to determine the degree of liver stiffness.
Perhaps it would useful that the authors develop a little more the contribution of these techniques to the diagnosis of non-alcoholic fatty liver disease.
It is a article with many references. Maybe too synthetic, but written for the understanding of any professional, from GP to gastroenterologist specialist. I don't have any others comments.
Thank you!
With consideration,
Magdalena Starcea
Author Response
We appreciate your comment. In this review article, we focused on NAFLD with emphasis on fat quantification rather than hepatic fibrosis. Since transient elastography and acoustic radiation force impulse (ARFI) elastography are suitable methods for assessing hepatic fibrosis, they are a bit far from the point of view we wanted to review. If given the opportunity, we would like to write a review article focusing on hepatic fibrosis in NAFLD patients. Thank you.
Reviewer 2 Report
In this review article, Jang and the colleagues summarized the recent development of liver fat evaluation method. The authors summarized about the liver fat quantification using US, CT, and MRI. The manuscript well-summarized, but it lacked important descriptions those readers really want to know. In addition, I have some concerns remaining as followings.
Comments
1. The title seems to be all non-invasive methods for the liver fat quantification were summarized in this review. The authors only summarized some imaging method. Please change the title, or add other methods (e.g. some scoring systems such as hepatic steatosis index (HIS), fatty liver index (FLI)…).
2. The readers would like to know whether fat accumulation in liver would change patients’ prognosis or not. Many recent works demonstrate the significance of liver fat deposits on the prognosis of NAFLD patients (e.g. Toh JZK CGH 2022, Vilar-Gomez E Hepatology 2023, Tamaki N Gut 2022). What is the clinical significance of fat deposition in the liver? The authors should describe about this issue.
3. Please add the references described about the comparison of different modalities described in this review (e.g. Jang JK Radiology 2022, Jung J Radiology 2022, Imajo K CGH 2022). Which modality is currently the most accurate? Please describe and tabulate the pros and cons of each modality in more detail.
no
Author Response
2-1. The title seems to be all non-invasive methods for the liver fat quantification were summarized in this review. The authors only summarized some imaging method. Please change the title, or add other methods (e.g. some scoring systems such as hepatic steatosis index (HIS), fatty liver index (FLI)…).
Answer) Thank you for your comment. Since we only dealt with imaging methods such as US, CT, and MRI for assessing hepatic steatosis, we used ‘Non-invasive imaging methods to evaluate NAFLD with fat quantification: A Review’ as the title of the current review article.
2-2. The readers would like to know whether fat accumulation in liver would change patients’ prognosis or not. Many recent works demonstrate the significance of liver fat deposits on the prognosis of NAFLD patients (e.g. Toh JZK CGH 2022, Vilar-Gomez E Hepatology 2023, Tamaki N Gut 2022). What is the clinical significance of fat deposition in the liver? The authors should describe about this issue.
Answer) Thank you for your helpful comment. According to your suggestion, we added the above mentioned references to describe the clinical significance of liver fat deposits on the prognosis of NAFLD patients.
2-3. Please add the references described about the comparison of different modalities described in this review (e.g. Jang JK Radiology 2022, Jung J Radiology 2022, Imajo K CGH 2022). Which modality is currently the most accurate? Please describe and tabulate the pros and cons of each modality in more detail.
Answer) We appreciate your comment. We added the references and supplementary explanation of US and MRI. In addition, we’ve mentioned that MRI-PDFF is the most accurate imaging method.
Round 2
Reviewer 2 Report
The authors well-revised their manuscript. I have no further comments on this manuscript.